# Beyond segmentation: an uncertainty-aware, end-to-end approach to functional lung image quantification

**Joshua R Astley**[1,2]  ID                                J.R.ASTLEY@SHEFFIELD.AC.UK
[1]*POLARIS, School of Medicine and Population Health, University of Sheffield, Sheffield, UK*
[2]*Insigneo Institute, University of Sheffield, Sheffield, UK*

**Helen Marshall**[1,2]                                     H.MARSHALL@SHEFFIELD.AC.UK
**Laurie J Smith**[1]                                       LAURIE.SMITH@SHEFFIELD.AC.UK
**Alberto M Biancardi**[1,2]                               A.BIANCARDI@SHEFFIELD.AC.UK
**Guilhem J Collier**[1,2]                                  G.J.COLLIER@SHEFFIELD.AC.UK
**Laura C Saunders**[1,2]                                  L.SAUNDERS@SHEFFIELD.AC.UK
**Jim M Wild**[1,2]                                          J.M.WILD@SHEFFIELD.AC.UK
**Bilal A Tahir**[1,2]                                        B.TAHIR@SHEFFIELD.AC.UK

**Editors:** Accepted for publication at MIDL 2025

## Abstract

Functional lung imaging modalities, such as hyperpolarized gas MRI, facilitate the visualization and quantification of regional lung ventilation. The ventilation defect percentage (VDP) is a highly-sensitive biomarker for quantifying small changes in lung function, derived from spatially co-registered functional hyperpolarized Xenon-129 ($^{129}$Xe)-MRI and structural proton ($^{1}$H)-MRI. However, manual-editing associated with segmentation-based workflows represents a time-consuming obstacle to delivering functional lung MRI results to clinicians. End-to-end deep learning (DL), which predicts final outputs without intermediary steps, frequently demonstrates improved performance on computer vision tasks; however, intermediary steps can no longer be interrogated. In this work, we developed the first end-to-end, uncertainty-aware DL framework for directly predicting VDP and its associated confidence. The direct prediction of VDP can potentially provide clinicians with important clinical data faster than segmentation-based methods.

**Keywords:** Functional imaging, lung, multi-modal, MRI, pulmonary, uncertainty-aware.

## 1. Introduction

Pulmonary imaging represents a key component in the clinical workflow for many patients with respiratory diseases, aiding in the diagnosis, monitoring, and treatment of these patients. Hyperpolarized gas MRI is a functional lung imaging modality that provides regional ventilation information with high spatial and temporal resolution (Stewart et al., 2021). Spatially co-registered functional hyperpolarized $^{129}$Xe-MRI and structural proton ($^{1}$H)-MRI scans facilitate the computation of biomarkers, such as the ventilation defect percentage (VDP); the VDP gives a percentage of defective lung region in comparison to the overall lung parenchymal volume (Stewart et al., 2021). To compute VDP, the segmentation of the ventilated lung and the lung cavity is required.

End-to-end deep learning (DL), which directly predicts the final output without intermediary steps, has demonstrated improved performance on computer vision tasks (Donahue

et al., 2020). We hypothesized that directly predicting VDP, without manual editing and segmentation steps, may improve the speed, accuracy and reliability of the predictions. However, end-to-end DL approaches inherently lack interpretability. Therefore, to improve trust in end-to-end DL, techniques can be employed which characterize the uncertainty of predictions, providing additional information above that of just the black-box prediction. Monte-Carlo (MC) dropout has been proposed as a method to determine the epistemic uncertainty of DL models by generating parameter space distributions of prediction uncertainty (Gal and Ghahramani, 2016). In this work, we develop an uncertainty-aware, dual-channel CNN-based framework to directly predict a key functional lung imaging metric, namely, VDP, from multi-modal $^{129}$Xe-MRI and $^{1}$H-MRI volumetric scans.

## 2. Methods

The dataset contained a total of 574 corresponding $^{1}$H-MRI and $^{129}$Xe-MRI scans from 47 healthy participants and 527 patients with a range of pulmonary pathologies. Patient data used in this work was pooled retrospectively from prospective clinical imaging studies and clinical referral cases. All participants underwent 3D volumetric $^{129}$Xe-MRI and $^{1}$H-MRI acquired in the coronal plane at approximately functional residual capacity + inhaled bag volume with full lung coverage at 1.5T (Stewart et al., 2018).

Ground-truth VDP values are calculated from the segmentation of the ventilated lung in $^{129}$Xe-MRI scans and the lung cavity estimations (LCEs) derived from the similar-breath structural $^{1}$H-MRI scan. LCEs, which segment the lung cavity in the spatial domain of $^{129}$Xe-MRI, were semi-automatically generated using several approaches (Astley et al., 2022a) (Hughes et al., 2018). $^{129}$Xe-MRI ventilated lung segmentations were generated from one of three independent methods, namely, SFCM (Hughes et al., 2018), linear-binning or CNN-based segmentation (Astley et al., 2022b). LCEs and $^{129}$Xe-MRI ventilated lung segmentations were subsequently manually edited by a wide selection of expert observers.

A dual-channel 3D CNN with a DenseNet-based (Huang et al., 2017) architecture was developed to predict a VDP value from paired $^{1}$H-MRI and $^{129}$Xe-MRI scans. Structural and functional imaging modalities were concatenated in a channel-wise fashion. All scans were normalized prior to network input and resized to 256x256x20 voxels. During network training, data augmentation was employed to minimize overfitting and improve robustness; affine and elastic rotation and scaling augmentations were employed at a probability of 0.1 and 0.5, respectively. The network was trained with a PReLU activation function, AdamW optimization (a=5x10$^{-5}$), and a smooth L1 loss (b=0.7) function. A batch size of 1 and a dropout rate of 0.15 was used. A data split of 80:10:10% was utilized, resulting in the following allocations: training (n=458), validation (n=58) and testing (n=58).

Epistemic uncertainty was determined via MC dropout and represents an Bayesian approximation of the Gaussian process. Posterior distributions, representing the epistemic uncertainty of the network, were constructed from 20 MC dropout iterations. K-means clustering was performed to classify these distributions into relative confidence classes.

To evaluate VDP predictions, the MAE was used as the primary metric. Normality was determined via Shapiro-Wilks tests and the appropriate parametric or non-parametric test was conducted to compare groups. A p-value <0.05 was considered statistically significant.

## 3. Results

A comparison between end-to-end DL predicted VDP values and those derived from a conventional segmentation-based workflow indicated no significant difference in VDP values, achieving a median (range) MAE VDP of 1.01% (0.002, 7.58) VDP across 58 testing cases (p=0.3677). A minimal Bland-Altman bias of -0.18 and a Pearson correlation of 0.94 was observed in the end-to-end DL VDP values; however, a few patients exhibited large VDP differences. Example readouts for six participants in the dataset are shown in Figure 1. Details are available on GitHub (https://github.com/POLARIS-Sheffield/Direct-VDP-Prediction).

Based on 20 MC dropout repeats, four confidence groups were constructed (very confident, confident, unconfident and very unconfident); significant differences in MAE were observed between the confident and unconfident metacategories (p=0.039).

## 4. Discussion

This novel dual-channel, end-to-end DL network exhibited no statistically significant differences compared to conventional segmentation-based workflows. A minimal bias was exhibited in Bland-Altman analysis, as well as a high correlation between segmentation-derived and end-to-end DL VDP values. The proposed end-to-end DL approach facilitates a more streamlined ventilation analysis workflow, using uncertainty-based confidence groups to stratify patients requiring manual intervention. The direct prediction of VDP can potentially provide clinicians with important clinical data faster than segmentation-based methods, allowing for the possibility of real-time, on-scanner, instant analysis.

One limitation of the uncertainty quantification technique deployed in our study is that some proportion of the quantified uncertainty is potentially due to differences in the raw VDP value; for example, patients with large VDPs may have larger uncertainty.

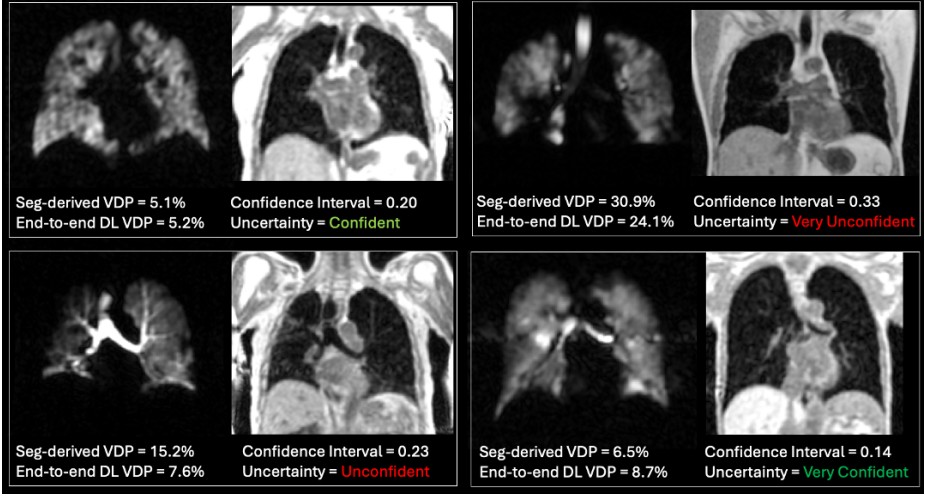

Figure 1: Example readouts with $^{129}$Xe-MRI and $^{1}$H-MRI scans, segmentation-derived and end-to-end DL VDP values, confidence intervals and confidence classes.

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
