# OpenReview forum: "Beyond segmentation: an uncertainty-aware, end-to-end approach to functional lung image quantification"
_MIDL.io/2025/Short_Papers — MIDL 2025 - Short Papers_

### Official Review · Reviewer_Ndbk · 2025-04-29

**Rating:** 3
**Confidence:** 4

**Summary:**

The paper presents a dual-channel CNN framework with MC dropout-based uncertainty estimation that directly predicts a key functional lung imaging metric—ventilation defect percent (VDP), using multi-modal volumetric scans from ¹²⁹Xe-MRI and ¹H-MRI.

**Strengths:**

*  Interesting paper with the focus on developing an uncertainty-aware deep learning framework for directly predicting lung ventilation defect percentage, bypassing intermediate manual editing and segmentation steps
* Reported results and analyses demonstrate the potential of the proposed approach

**Weaknesses:**

* Some details of the dataset are missing, such as source, pathologies, VDP ranges, etc.
* Several segmentation approaches are mentioned for the ground truth VDP calculation. I wonder if the choice of segmentation has any major impact on the ground truth VDP as well as the proposed direct prediction performance.
* It would be interesting to compare the proposed end-to-end VDP prediction against a two-stage (segmentation+VDP) approach; could be done by training two separate CNNs.
* The backbone CNN architecture could be replaced by a more recent one if there is any impact on the model performance.

---

### Decision · Program_Chairs · 2025-05-01

Accept